# What Is the Value of Ultrasound in Individuals ‘At-Risk’ of Rheumatoid Arthritis Who Do Not Have Clinical Synovitis?

**DOI:** 10.3390/healthcare9060752

**Published:** 2021-06-18

**Authors:** Andrea Di Matteo, Davide Corradini, Kulveer Mankia

**Affiliations:** 1Rheumatology Unit, Department of Clinical and Molecular Sciences, Polytechnic University of Marche, “Carlo Urbani” Hospital, Jesi, 60035 Ancona, Italy; 2Leeds Institute of Rheumatic and Musculoskeletal Medicine, University of Leeds, Leeds LS74SA, UK; K.S.Mankia@leeds.ac.uk; 3Rheumatology Unit, University of Cagliari and AOU University Clinic of Cagliari, 09042 Monserrato, Italy; corradini.davide94@gmail.com; 4National Institute for Health Research Leeds Biomedical Research Centre, Leeds Teaching Hospitals NHS Trust, Leeds LS74SA, UK

**Keywords:** musculoskeletal ultrasound, power Doppler signal, synovial hypertrophy, prediction, ‘at-risk’, rheumatoid arthritis, inflammatory arthritis

## Abstract

The identification of biomarkers that help identify individuals at imminent risk of progression to rheumatoid arthritis (RA) is of crucial importance for disease prevention. In recent years, several studies have highlighted the value of musculoskeletal (MSK) ultrasound (US) in predicting progression to inflammatory arthritis (IA) in individuals ‘at-risk’ of RA. These studies have highlighted the following main aspects: first, in RA-related autoantibody-positive individuals, MSK symptoms seem to develop before ‘sub-clinical’ joint inflammation occurs on US. Second, the detection of ‘sub-clinical’ synovitis (and/or bone erosions) greatly increases the risk of IA development in these ‘at-risk’ individuals. US has a potential key role for better understanding the ‘pre-clinical’ stages in individuals ‘at-risk’ of RA, and for the early identification of those individuals at high risk of developing IA. Further research is needed to address questions on image analysis and standardization. In this review, we provide an overview of the most relevant studies which have investigated the value of US in the prediction of RA development in individuals ‘at-risk’ of RA who have MSK symptoms, but no clinical evidence of IA. We highlight recent insights, limitations, and future perspectives of US use in this important population.

## 1. Introduction

Early rheumatoid arthritis (RA) is now considered a disease ‘continuum’ rather than a fixed phenotype [1,2]. The earliest stages of this ‘continuum’, stages A and B as per the definition of the European League Against Rheumatism (EULAR) Standing Committee on Investigative Rheumatology (Figure 1), encompass individuals with genetic predisposition and/or environmental risk factors for RA (e.g., first-degree relative of RA probands, presence of shared epitope and/or cigarette smoking) [3]. Some of these ‘at-risk’ individuals will develop systemic autoimmunity and RA-related autoantibodies (stage C), such as anti-citrullinated protein antibody (ACPA) and/or rheumatoid factor (RF). Later musculoskeletal (MSK) symptoms (stage D) can occur, which may represent the initial signs of a transition from autoimmunity to inflammation [4]. The occurrence of systemic autoimmunity is an important pathological step (i.e., the ‘first hit’ in RA pathogenesis) which drives progression to clinical disease [5]. In fact, ACPA can be detected in patients’ sera years before the onset of RA and, when present, they significantly increase the risk of progression to RA [6,7,8,9,10]. However, progression to undifferentiated/unclassifiable arthritis or RA (stage E) is not inevitable; some ‘at-risk’ individuals (i.e., with RA-related Ab and/or genetic/environmental risk factors for RA) will remain healthy and will never develop arthritis.

The identification of biomarkers that can help identify individuals at high risk of future RA is of utmost importance for disease prevention [11]. Such biomarkers enable risk-stratification and, therefore, inform the practical management of these ‘at-risk’ individuals: those who are at high risk should be followed carefully and may be considered for therapeutic trials for arthritis prevention. On the other hand, those at low risk may be followed less intensively and re-evaluated only if symptoms change.

In recent years, there has been a growing evidence base supporting the value of MSK ultrasound (US) in ‘at-risk’ individuals across the different stages of the early RA ‘continuum’. US appears to be particularly valuable in those individuals with MSK symptoms, but without clinical arthritis (i.e., stage D), and in those with undifferentiated (i.e., not fulfilling the RA criteria) arthritis (i.e., stage E) [12,13]. Indeed, the prognostic role of US-detected joint or tendon inflammation in identifying persistent disease in patients with early undifferentiated arthritis has been highlighted by several studies [14,15,16,17]. The results of these studies have several important implications for the early diagnosis and treatment (i.e., ‘window of opportunity’) of RA patients, especially in those who are seronegative for ACPA and/or RF [18,19].

Another important population, which will be the focus of this narrative review, is individuals ‘at-risk’ of RA who have MSK symptoms, but no clinical evidence of inflammatory arthritis (IA). There is growing interest in this population for arthritis prevention studies. We will provide an overview of the most relevant studies which have evaluated the role of US in the prediction of RA development in such individuals. We will highlight recent insights, limitations, and future perspectives of US use in this pre-arthritis population.

## 2. The Role of Ultrasound (US) in Predicting Inflammatory Arthritis in ‘At-Risk’ Individuals

In recent years, several US and/or magnetic resonance imaging (MRI) studies have identified a ‘new’ population which represents an important stage in the early RA ‘continuum’. This population, which sits between stage D and stage E of the RA ‘continuum’, is represented by ‘at-risk’ individuals with imaging evidence of sub-clinical inflammation (and/or joint damage) but no clinical arthritis [20,21]. Figure 2 shows the most representative US abnormalities detectable in these individuals: synovitis, tenosynovitis and bone erosions.

The first study evaluating the role of US in predicting arthritis in ‘at-risk’ individuals with arthralgia, but without clinical synovitis, was carried out by the Amsterdam group [22]. One hundred and ninety-two ‘at-risk’ individuals with arthralgia and positive ACPA and/or RF underwent a US scan of the joints. However, the US dataset was limited and varied between individuals; only joints tender on physical examination (and/or reported as painful in the patient’s medical history) or, in the case of metacarpophalangeal (MCP) joints, proximal interphalangeal (PIP) joints and metatarsophalangeal (MTP) joints, the directly adjacent and contralateral joints to the tender joints, were imaged. Joint effusion, synovial hypertrophy, power Doppler (PD) signal and tenosynovitis were scored according to a semi-quantitative scale (0–4). Grades ≥ 2 of joint effusion, synovial hypertrophy and tenosynovitis and grades ≥ 1 of PD signal were considered as pathological. While the presence of joint effusion, synovial hypertrophy and PD signal in a joint was associated with arthritis development in that joint [odds ratio (OR) 3.07 (95% confidence interval (CI) 1.05–8.94), OR 5.45 (95% CI 2.32–12.8), and OR 5.50 (95% CI 2.57–11.9), respectively] only a positive trend between these US findings and the development of IA was observed at patient level, and this did not reach statistical significance [OR 2.05 (95% CI 0.80–5.27), OR 1.41 (95% CI 0.54–3.65), OR 1.54 (95% CI 0.67–3.54), respectively].

In 2014, 100 patients with new non-specific MSK symptoms and a positive anti-cyclic citrullinated peptide (anti-CCP) antibody test from the ‘Leeds CCP study’ were included in a prospective study to develop a model to predict IA development [23]. The model included four variables: tenderness of the hands or feet, early morning stiffness duration ≥30 min, high level anti-CCP antibody, and intra-articular PD signal on US. Using an US scanning protocol which evaluated the wrists, MCP joints, and PIP joints, PD signal was detected in 33 out of 100 (33%) ‘at-risk’ individuals. In the univariable analysis, PD signal showed a hazard ratio (HR) of 1.88 (95% CI 1.07–3.29) for the development of IA. This predictive value was also confirmed in the multivariable Cox regression analysis with a HR of 1.84 (95% CI 1.04–3.27, *p* = 0.037), and with a HR of 1.92 (95% CI 1.06–3.50) when HLA-DR shared epitope was also included in the model. The discrepant results between the Amsterdam and Leeds studies may be explained by the different risk profiles of the ‘at-risk’ individuals, as well as the different US protocols. All individuals in the Leeds study had positive anti-CCP antibodies and carried a higher risk of developing clinical arthritis (more than 40% developed IA at median 8.6 months). In the Amsterdam cohort, only 70% of at-risk individuals had positive ACPA and the rate of progression to IA at 12 months was lower (31%). Moreover, the US protocol in the Leeds CCP study evaluated a larger and more comprehensive set of joints which also included non-tender joints on physical examination. On the other hand, only tender joints on physical examination (± adjacent and contralateral to tender joints in case of MCP, PIP and MTP joints) were imaged in the Amsterdam study. It is possible that subclinical inflammation in other non-imaged joints may have been missed.

Subsequently, a larger and more comprehensive US study was performed by the Leeds group; the role of US in predicting IA development was investigated in 136 ‘at-risk’ individuals from the Leeds cohort [24]. The association between grey scale (GS) synovitis, PD signal, bone erosions and clinical arthritis development was evaluated both at patient and joint level. Synovitis and bone erosions were defined according to the Outcome Measures in Rheumatoid Arthritis Clinical Trials (OMERACT) definitions [25]. In addition, GS and PD findings were scored using a semi-quantitative method (0–3) using the EULAR-OMERACT scoring system [26]. The US scanning protocol included 32 joints [wrists, MCP joints, PIP joints, metatarsophalangeal (MTP) joints]. In this study, GS synovitis ≥1, PD signal ≥1 and bone erosions were found in 72.8% (99/136), 25.7% (35/136) and 5.9% (8/136) of ‘at-risk’ individuals, respectively. Of the individuals who developed IA, 86% had at least one US abnormality at baseline. In this study, rate of progression to clinical arthritis (both at patient and joint level) was significantly higher in ‘at-risk’ individuals with US abnormalities (i.e., GS synovitis or PD positive synovitis or bone erosion) than in those without baseline US abnormalities. The highest HR for IA development at patient level was observed for PD signal ≥ 2 [HR 3.7 (95% CI 1.7–6.5), *p* < 0.001). In addition, the presence of PD in a joint (any score >0) was associated with a 10-fold increase in risk of developing clinical synovitis in that joint [HR 10.3 (95% CI 5.9–18.2, *p* < 0.001).

Zufferey et al. evaluated the predictive value of US for future RA development in 80 consecutive patients with inflammatory arthralgia lasting > 6 weeks, without clinical synovitis, and seronegative for anti-CCP and RF [27]. Therefore, unlike the Amsterdam and Leeds at-risk cohorts, all individuals included in this study had negative RA-related autoantibodies. The US protocol was performed according to the Swiss Sonography Group in Arthritis and Rheumatism (SONAR) score, which evaluates the same joints as the DAS-28, but excludes the thumbs and shoulders, and uses cut-offs for ‘active’ inflammatory arthritis based on the grade of B-mode synovitis and PD signal (quantified according to the semi-quantitative scoring system proposed by OMERACT) and the number of joints with US pathological findings [28]. However, in this study, PD signal was not included in the prediction analyses due to its low prevalence (5%) in the population. Twenty out of 80 (25%) patients had a positive SONAR score (≥2 joints with at least grade 2 synovitis) at baseline and this was associated with future RA (or IA) development in the multivariate analysis [OR 10.1 (95% CI 1.1–49)]. Interestingly, the negative predictive value for IA/RA development when no B-mode synovitis was found was 94%.

In a study by van der Ven et al., the predictive value of US for IA development within 1 year was studied in 159 patients with inflammatory arthralgia, without clinical synovitis, but with or without RA-related autoantibodies (ACPA-positive 15%; RF-positive 26.4%) [29]. Therefore, this can be regarded as an intermediate population between the one in the study by Zufferey et al. [27] and the Leeds and Amsterdam cohorts [22,23,24]. The patients underwent a 26-joint US protocol evaluating the wrists, MCP joints 2–5, PIP joints 2–5 and MTP joints 2–5. GS synovitis and PD signal were scored according to a semi-quantitative scoring system (0–3) following the recommendations of the Spanish Society of Rheumatology, which uses a modified version of the OMERACT definitions of US pathology. US synovitis, defined as GS synovitis grade 2 or 3 and/or presence of PD signal, was found in 17 out of 31 (59%) patients who developed IA and in 44 out of 143 (32%) who did not (*p* = 0.007). At 1 year follow-up, only 16% of individuals developed IA. PD signal was found in 9 out of 31 (31%) patients who progressed to IA and in 17 out of 143 (12%) who did not (*p* = 0.012). In this study, the sensitivity and specificity of US synovitis for future IA development were 59% and 68%, respectively. Interestingly, absence of US synovitis had a negative predictive value for future IA development of 89%. In the multivariable analysis, the presence of US synovitis or PD signal in ≥1 joint was significantly associated with future IA development [OR 3.03 (95% CI 1.69–5.41) and OR 3.12 (95% CI 1.61–6.03), respectively].

Following the original study by van de Stadt et al., the Amsterdam group evaluated a new cohort of 163 seropositive patients (ACPA negative/RF positive 44%; ACPA positive/RF negative 27%; ACPA positive/RF positive 29%) with arthralgia but without clinical arthritis [30]. In this study, the authors performed an US scanning protocol evaluating a standardized set of joints (regardless of joint symptoms); this included wrists, MCP2 and MCP3 joints, PIP2 and PIP3 joints and MTP2, MTP3, and MTP5 joints. Synovial thickening and PD signal were scored using a semi-quantitative scale [0–3 according to Szkudlarek [31]] and considered as abnormal if synovial thickening ≥ grade 2 and PD signal ≥grade 1. Baseline US synovial thickening, which was found in 49 out of 163 (30%) individuals, was predictive for IA development at patient level in the multivariable analysis [OR 6.1 (95% CI 1.6–23.2, *p* < 0.01). PD was unfrequently detected (4% of patients) and was not associated with the development of IA at patient level [OR 1.7 (0.3–10.2), *p* = 0.55]. The very low prevalence of PD signal detected in this population is somewhat surprising. Indeed, this prevalence was lower than the one observed in the study van der Ven et al. which also included seronegative patients (29). In addition, the rate of progression to IA at 12 months follow-up was higher in this study than in that by van der Ven et al. (31% vs. 16%, respectively). A possible explanation could be represented by the different US scanning protocols used in the two studies, which was larger and more comprehensive in the latter. Consequently, subclinical synovitis on US was less likely to be missed. Moreover, while the pre-requisite for including patients in this study was having ‘arthralgia’, patients in the study by van der Ven et al. had more ‘inflammatory’ characteristics, such as at least two painful joints in hands, feet or shoulders, plus two of the following: early morning stiffness duration > 60 min, inability to make a fist in the morning, pain when shaking someone’s hand, pins and needles in the hands, difficulties wearing rings or shoes, family history of RA and/or unexplained fatigue for <1 year.

In addition to the presence of ‘sub-clinical’ inflammation, the detection of joint structural damage (i.e., bone erosions) on US appears also to be associated with IA development in ‘at-risk’ individuals with MSK symptoms but without clinical synovitis.

In the Leeds cohort, US detected bone erosions were observed in 30 out of 136 (22%) ‘at-risk’ individuals with MSK symptoms, but without clinical arthritis. US bone erosions were predictive for IA development with an OR of 2.9 (95% CI 1.7 to 5.1), *p* < 0.001 [24]. In a larger subsequent study of 419 at-risk individuals from the same cohort, the prevalence and distribution of US bone erosions at the classical sites for RA damage (MCP2 and MCP5 joints, and MTP5 joints), and their association with development of clinical arthritis, was investigated [32]. Bone erosions, identified according to the OMERACT definitions, were detected in ≥1 joint in 41/419 subjects (9.8%); these were associated with progression to IA, and its timing, at patient level. The risk of progression was highest for the following: bone erosions in >1 joint [OR 10.6 (95% CI 1.9–60.4, *p* < 0.01)] and bone erosions and synovitis in ≥1 MTP5 joint [OR 5.1 (95% CI 1.4–18.9), *p* = 0.02]. Interestingly, in a more recent study by the same investigators, the prevalence of conventional radiography (CR) detected bone erosions in the Leeds cohort was lower (17/418, 4.1%) and these were not associated with evolution to IA [33]. These results suggest that US may represent the optimal imaging technique in the assessment of bone erosions in ‘at-risk’ individuals without clinical arthritis, due to its higher sensitivity in comparison with CR, especially in the early stages of the disease.

## 3. Discussion

In this non-systematic literature review, we analysed US studies which have evaluated the predictive value of US for progression to IA in ‘at-risk’ individuals with MSK symptoms, but without clinical arthritis. The main general and US characteristics of these studies are reported in Table 1 and Table 2, respectively.

These studies have highlighted the following main aspects: first, only a minority of ‘at-risk’ individuals with MSK symptoms have US abnormalities at baseline (i.e., first visit) thus suggesting that, in most autoantibody-positive individuals, MSK symptoms develop before sub-clinical joint inflammation occurs on imaging. As shown in Table 2, the prevalence of PD signal ranged from 4% to 33% in the different studies. Second, the detection of baseline GS or PD positive synovitis (and/or bone erosions) significantly increases the risk of that individual developing IA.

The detection of ‘sub-clinical’ inflammation on US has potential implications for how clinicians manage at-risk individuals. Indeed, in a recent UK survey, almost 75% of rheumatologists said that they would start treatment with a disease modifying anti-rheumatic drug (DMARD) in patients with positive ACPA, MSK symptoms and US-detected PD signal, in the absence of any clinical synovitis [34]. Moreover, some might argue that US ‘sub-clinical’ inflammation might represent a late stage in the RA ‘continuum’, and when present may represent an irreversible progression to IA. Indeed, recent data suggest that the presence of PD in more the three joints, in anti-CCP ‘at-risk’ individuals, is virtually synonymous with imminent progression to RA [35]. In the context of clinical trials for disease prevention, this poses the question whether ‘at-risk’ individuals should be approached for these trials before the development of ‘sub-clinical’ synovitis. In other words, if treating these patients before the occurrence of US ‘sub-clinical’ synovitis (potentially representing the ‘second hit’ in RA pathogenesis) would prevent the development of any joint inflammation, and therefore produce better outcomes.

Most of the studies presented in this review explored the predictive value of US findings in multivariable models also including clinical and serological factors, such as tenderness in the hands or feet, early morning stiffness, or RA-related antibodies [23,27,29,30,32]. This suggests that US has an additional value when used alongside other clinical or serological variables. Clearly, ACPA positivity is a well-known strong risk factor for RA development, and it has been used to define ‘at-risk’ status in several studies. Likewise, most of ACPA positive individuals (especially those with low titer autoantibodies) will never develop RA [36]. In this context, US has a promising role in the identification of those individuals who are at especially high risk of developing RA, with potential implications on risk stratification and therefore management of these individuals.

On the other hand, recent data show that progression to IA is not inevitable even in those patients with RA-related systemic autoimmunity, MSK symptoms, and ‘sub-clinical’ inflammation on US. Indeed, the specificity of sub-clinical inflammation on US for IA development has been questioned by a very recent study by Rogier et al. [37]. The authors re-analysed US findings from two Dutch cohorts of patients with arthralgia (but without clinical synovitis) described above [29,30] to investigate the frequency of progression to IA at 1 year.

In the two cohorts, 54% and 44% of the individuals with ACPA and ‘sub-clinical’ synovitis on imaging did not progress to IA at 1 year follow-up. In addition, in the ACPA-negative groups, only 34% and 15% of individuals with sub-clinical synovitis developed IA. The authors suggested that using ‘sub-clinical’ synovitis to identify patients who will develop RA introduces a high risk of ‘false-positive’ results, thus potentially leading to overtreatment.

The different rate of progression to IA in ‘at-risk’ individuals with sub-clinical synovitis identified in the studies described above may depend on several important factors. Many of them relate to qualitative nuances in the data provided by the US: the type of US pathological findings (i.e., GS synovial hypertrophy, synovial effusion, PD signal, bone erosions, isolated or in combination), the grade of US abnormality (present/absent or semi-quantitative scoring which distinguishes between different grades of severity), the distribution of the US pathological findings (joints, tendons, RA-specific sites, such as MCP2, MCP5, MTP5 or ulnar styloid), the number of joints with US sub-clinical ‘inflammation’, and the target population (‘at-risk’ individuals with MSK symptoms and positive RA-related antibodies vs. seronegative, inflammatory arthralgia vs. unspecific MSK symptoms). The huge variability in the methodology of the included studies (e.g., discrepancies in US protocols and equipment used, heterogeneity in the US abnormality definitions adopted) may explain the different predictive values of US pathological findings in ‘at-risk’ individuals, both at joint and patient level. Indeed, these studies are remarkably heterogeneous. The most relevant differences can be observed regarding eligibility criteria, including symptoms and serologic profile, and US scanning protocol. Therefore, despite most studies highlighting the great utility of US in improving prediction to IA in at-risk individuals, the interpretation of the relevance and validity of the US findings remain partially undefined.

Interestingly, while the great majority of studies have evaluated US inflammatory findings at joint level, only one has evaluated the prevalence and predictive role for IA development of tenosynovitis [22]. In a study on 107 early arthritis patients with clinical synovitis of short duration (<3 months), US tenosynovitis of the finger flexor tendons was independently associated with evolution to RA [16]. Interestingly, MRI tenosynovitis showed the greatest association with IA development in 150 patients with clinically suspect arthralgia (HR 7.56, 95% CI 3.30, 17.32, *p* < 0.001) [38]. Further investigations in this regard may be warranted. In addition, studies which have investigated US in at-risk individuals have exclusively reported on a single baseline assessment. Whether and how US abnormalities change, or develop over time, is not clear.

Most studies conducted in ‘at-risk’ populations have adopted comprehensive US scanning protocols evaluating a large number of joints. This is feasible in a research setting but perhaps challenging and time-consuming in routine clinical practice. Moreover, US pathological findings have also been found in healthy subjects, especially GS changes in the foot joints [39]. In addition, inflammation may occur with structural changes of osteoarthritis in small and large joints [40]. Therefore, it would appear relevant to not include all US changes in the analysis, some of which may be non-pathological and therefore not expected to predict IA, and to interpret the US findings always in the context of other joint findings (i.e., concomitant osteoarthritis). In this regard, it should be noted that only two US studies included a control group (Table 1). As also shown by previous MRI studies on ‘at-risk’ cohorts, the use of a pathological ‘cut-off’ would intuitively increase the specificity of the US findings [41,42]. Importantly, the EULAR-OMERACT US group has recently produced new definitions and quantification system for RA synovitis, including combination and individual synovial hypertrophy and PD components [43]. Further studies are needed to explore the diagnostic performances of these definitions in patients at risk without clinical synovitis. We have demonstrated that a targeted US protocol, focused on the classical sites for RA damage (MCP2, MCP5 and MTP5 joints) is useful to predict progression (and its timing) to IA in CCP+ ‘at-risk’ individuals, thus improving risk-stratification and informing the management of these individuals [32]. Further research is needed to identify the ‘ideal’ scanning protocol, with the optimal predictive accuracy, which could also be applied in real-life clinical settings.

The lifetime costs of a patient with RA are known to be substantial [44,45]. Prediction and (possible) prevention of RA has the potential to reduce costs in several ways. However, this hypothesis, and the role of US in this context, need to be corroborated in health economics studies.

In addition, more investigations are needed to evaluate the impact of the US findings on the management of ‘at-risk’ individuals, and if changing this according to the presence of the US findings would produce better outcomes in the long-term. Interestingly, the very small number of clinical trials on diseases prevention in ‘at-risk’ individuals have not adopted US as mandatory inclusion criteria nor have they included it in the outcomes [46].

## 4. Conclusions

In conclusion, US has a promising role for the early identification of those individuals at high risk of imminently developing IA and for better understanding of the different ‘pre-clinical’ stages in individuals ‘at-risk’ of RA. Technical considerations, such as image analysis and protocol standardization, are likely to be an important explanation for discrepancies between studies. This remains an issue for further investigation.

## Figures and Tables

**Figure 1 healthcare-09-00752-f001:**
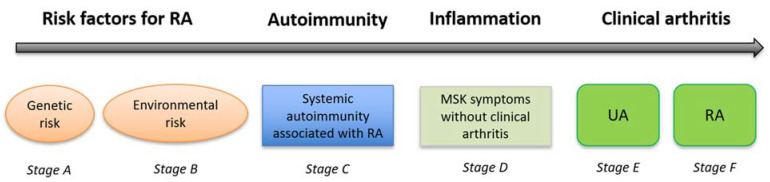
Overview of the pre-clinical phases of RA. Stages along the RA ‘continuum’ defined by the European League Against Rheumatism (EULAR) Standing Committee on Investigative Rheumatology. Legend: MSK: musculoskeletal; RA: rheumatoid arthritis, UA: undifferentiated arthritis.

**Figure 2 healthcare-09-00752-f002:**
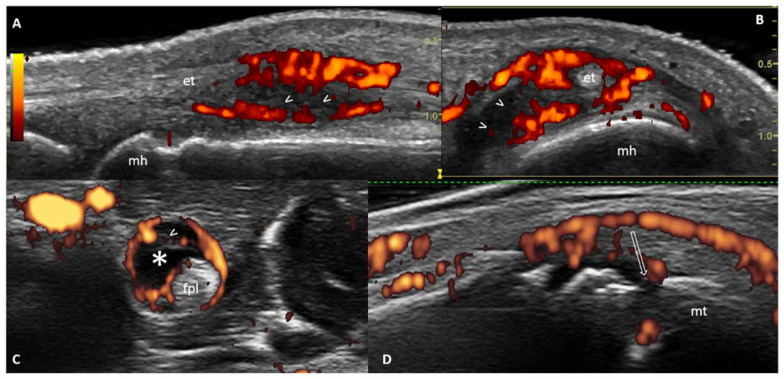
Representative ultrasound (US) findings in at-risk individuals with musculoskeletal symptoms but without clinical arthritis. Longitudinal (**A**) and transverse (**B**) US scan of the dorsal aspect of the 3rd metacarpophalangeal joint. US shows the presence of “active” synovitis, characterized by synovial hypertrophy (arrowheads) with PD signal (red spots) which also envelops the finger extensor tendon (et). (**C**) Transverse US scan of the flexor pollici longus tendon shows “active” tenosynovitis with synovial effusion (asterisk) and synovial hypertrophy (arrowhead) within the tendon sheath (asterisk) surrounded by PD signal (red spots). (**D**) Longitudinal US scan of the lateral aspect of the 5th metatarsophalangeal joint. Note the presence of a bone erosion (empty arrow) filled with power Doppler signal (red spots). Legend: et = finger extensor tendon; fpl = flexor pollici longus; mh = metacarpal head; mt = metatarsal head. ((**A**) Di Matteo, personal images).

**Table 1 healthcare-09-00752-t001:** General main characteristics of the studies.

Authors	Year of Publication	Study Design	Number of ‘At-Risk’ Individuals	ACPA Positive	RF Positive	ACPA or RF Positive	MSK Involvement	Control Group	Proportion of Progressors to IA	Median Time of Progression to IA	Outcome
van de Stadt et al. [22]	2010	PS	192	69%	63%	100%	Arthralgia (i.e., non-traumatic pain in any joint)	Y (9 HC)	23%	Mean: 11 M (SD ± 9)	US findings associated with clinical arthritis development at joint level but not at patient level
Rakieh et al. [23]	2014	PS	100	100%	46%	100%	New onset MSK symptoms	N	50%	Median: 7.9 M (IQR 3.2, 14.5)	PD signal predictive for IA development in the multivariable analysis
Nam et al. [24]	2016	PS	136	100%	45%	100%	New onset MSK symptoms	Y (48 HC)	41.9%	Median: 18.3 M (range 0.1–79.6)	US findings (especially PD) predictive for progression to IA
Zufferey et al. [27]	2016	RS	80	0%	0%	0%	Polyarthralgia	N	8.7%	Median: 18 M	US only independent predictor to IA development in the multivariable analysis
van der Ven et al. [29]	2017	PS	159	15%	26%	NR	Inflammatory arthralgia	N	16%	1 year follow-up	PD in combination with clinical parameters is associated with IA development
van Beers-Tas et al. [30]	2018	PS	163	46%	73%	100%	Arthralgia	N	31%	Median 12 M (IQR 5–24)	US synovial thickening predicted IA development (excluding MTP joints). PD signal infrequent and not predictive
Di Matteo et al. [32]	2020	PS	419	100%	38%	100%	New onset MSK symptoms	N	30.7%	Median: 9.9 M (IQR 3.6–26.7)	BE in >1 joint, and BE in combination with US synovitis in the MTP5 joints, most predictive for the development of IA

Legend. ACPA: anti-citrullinated protein antibody; BE: bone erosions; CS: clinical synovitis; HC: healthy controls; IA: inflammatory arthritis; IQR: inter-quartile range; M: months; MSK: musculoskeletal; MTP: metatarsophalangeal joints; N: no; NR: not reported; PD: power Doppler; PS: prospective study; RF: rheumatoid factor; RS: retrospective study; SD: standard deviation; SY: synovial hypertrophy; US: ultrasound; Y: yes.

**Table 2 healthcare-09-00752-t002:** Main US-related characteristics of the studies (scanning protocols, definitions and type of US elementary lesions evaluated).

Authors	Sonographer Inter-Observer Agreement	Sonographer Blind to Clinical Data	Scanned Areas	Number of Joints Evaluated	US Probe Frequencies		US Elementary Lesions Assessed (Prevalence %)					US Definitions Used	Grading of the US Findings	Baseline or Longitudinal Scans
					GS (MHz)	PD (MHz)	SH	SE	PD	BE	TS			
van de Stadt et al. [22]	Y	Y	Painful, adjacent and contralateral joints	Variable for each patient	8–15	7.3–8.9	Y (12.5)	Y (11.4)	Y (17.1)	N	Y (6.7)	Szkudlarek	Semiquantitative	Baseline
Rakieh et al. [23]	Y	N	Bilateral wrists, 1–5 MCP and 1–5 PIP joints	22	8–15	NR	N	N	Y (33)	N	N	Naredo/Torp-Pedersen	Dichotomic	Baseline
Nam et al. [24]	Y	N	Bilateral wrists, 1–5 MCP, 1–5 PIP, 1–5 MTP joints	32 and other joints if symptomatic	*	*	Y (95.5)	N	Y (33)	Y (20.5)	N	OMERACT	Semiquantitative	Baseline
Zufferey et al. [27]	N	N	Bilateral elbows, wrists, 2–5 MCP, 2–5 PIP and knee joints	22	7–13	NR	Y (25)	N	N	N	N	SONAR	Semiquantitative	Baseline
van der Ven et al. [29]	N	Y	Bilateral wrists, 2–5 MCP, 2–5 PIP, 2–5 MTP joints	26	10–18	NR	N	Y (35.6)	Y (14.9)	N	N	OMERACT modified version by The Spanish Society of Rheumatology	Semiquantitative	Baseline
van Beers-Tas et al. [30]	Single operator	Y	Bilateral wrists, 2,3 MCP, 2,3 PIP, 2,3,5 MTP joints	16	11.4	8.9	Y (30)	N	Y (4)	N	N	Szkudlarek	Semiquantitative	Baseline
Di Matteo et al. [32]	Y	Y	Bilateral 2,5 MCP and 5 MTP joints	6	*	*	Y (NR)	N	Y (22.9)	Y (9.8)	N	EULAR-OMERACT	Dichotomic	Baseline

Legend. BE: bone erosion; EULAR: European League Against Rheumatism; GS: gray scale; MCP joints: metacarpophalangeal joints; MHz: megahertz; MTP joints: metatarsophalangeal joints, N: no; NR: not reported; OMERACT: Outcome Measures in Rheumatology; PIP joints: proximal interphalangeal joints; PD: power Doppler; SE: synovial effusion; SH: synovial hypertrophy; TS: tenosynovitis; US: ultrasound d; Y: yes. *: different US machines were used during the study.

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
