# Peer review of "What Is the Value of Ultrasound in Individuals ‘At-Risk’ of Rheumatoid Arthritis Who Do Not Have Clinical Synovitis?"

_healthcare, 2021, doi:10.3390/healthcare9060752_

Round 1
Reviewer 1 Report
This paper reviews the predictive value of US in individuals at risk of RA who have MSK symptoms.
It is not a systematic review and the authors have not described their search strategy
Many of the studies discussed confirm the strong predictive value of ACPA in determining progression to RA. They also showed that only a minority of individuals with MSK symptoms have US abnormalities at baseline and even in the presence of US detected synovitis only half progressed over the first year of follow-up
The unanswered questions seem to me to be:
Is there any evidence that US offers any benefit over clinical examination and ACPA in terms of predicting progression to RA?
Is there evidence that US facilitates earlier treatment and changes outcomes?
Is the significant time and cost of US worth it?
I am not sure that the Authors have evidenced their concluding statement ‘existing data supports the use of US for the early identification of those individuals at high risk of imminently developing IA’.
Author Response
This paper reviews the predictive value of US in individuals at risk of RA who have MSK symptoms. It is not a systematic review and the authors have not described their search strategy. Many of the studies discussed confirm the strong predictive value of ACPA in determining progression to RA. They also showed that only a minority of individuals with MSK symptoms have US abnormalities at baseline and even in the presence of US detected synovitis only half progressed over the first year of follow-up.
- Thanks to the Reviewer for their comments. The ‘narrative’ nature of this review has been now underlined in the text.
The unanswered questions seem to me to be:
Is there any evidence that US offers any benefit over clinical examination and ACPA in terms of predicting progression to RA?
- This point is well taken. Most of the studies presented in this review have explored the predictive value of US findings in multivariable models also including clinical and serological factors, such as tenderness in the hands or feet, EMS, and RA-related antibodies. This suggests that US has an additional value when used alongside other clinical or serological variables. Clearly, ACPA positivity is a well-known and strong risk factor for RA development. The presence of ACPA has been used to define ‘at-risk’ status in several studies. This has been already discussed in the introduction. However, it is also known that most ACPA positive individuals will remain healthy and never develop RA. In this context, US has a promising role in further risk stratifying ACPA positive individuals, identifying a subgroup who are at especially high risk of developing RA, with potential implications for the management of these individuals. A few sentences have now been added in the manuscript (page 9, lines 22-31).
Is there evidence that US facilitates earlier treatment and changes outcomes?
- This point is well taken. There is some preliminary evidence for undifferentiated arthritis (e.g. Salaffi et al., Clinical and Experimental Rheumatology 2010, Freeston et al., Annals of the Rheumatic Diseases 2010, Iqbal H, Rheumatology 2019) but not yet in individuals without synovitis. We believe that this will be the subject of future trials. However, as shown by a recent survey by Mankia et al., rheumatologists are already acting on US and treating earlier, even in absence of an evidence base. Clearly, this is an aspect that requires further investigations. A few sentences have been added in the manuscript (page 11, lines 1-5).
Is the significant time and cost of US worth it?
- The use of MSK US is already routine practice in many EACs as it refines diagnosis in early arthritis. We would extend to at risk phase where it is has shown the ability of further improving risk-stratification. In the light of (potential) disease prevention, we believe that scanning selected anti-CCP2+ individuals (i.e., those with higher titer CCP Ab and MSK symptoms) would not significantly raise the costs. Indeed, identifying and treating individuals at-risk before they develop the disease has the potential to significantly reduce long-life costs related to rheumatoid arthritis. However, this hypothesis has to be demonstrated by longitudinal studies. This point has been further discussed in the manuscript (page 10, last paragraph). The US scanning protocols which have been adopted in most studies evaluated an extensive number of joints. This could be time-consuming and therefore not suitable in clinical practice. As we stated in the discussion, we believe that further research is needed to “identify the ‘ideal’ scanning protocol, with the optimum predictive accuracy, which could also be applied in real-life clinical settings”.
I am not sure that the Authors have evidenced their concluding statement ‘existing data supports the use of US for the early identification of those individuals at high risk of imminently developing IA’.
- We have now changed the conclusions of the manuscript according to the Reviewers’ suggestions.
Reviewer 2 Report
The text is very clear and very informative on current and relevant subject.
In this review, the authors provided an overview of the most relevant studies that investigated the value of US in predicting the development of RA in individuals "at risk" for RA who have MSK symptoms, but no clinical evidence of Inflammatory arthritis.
They also highlighted the recent perceptions, limitations and future perspectives of the use of US in these patients. The text is very clear and very informative on current and relevant subject
Author Response
We would like to thank the reviewer for their kind comments.
Reviewer 3 Report
Comments to the Authors: Manuscript ID: healthcare-1247294
Title: “What is the value of ultrasound in individuals ‘at-risk’ of rheumatoid arthritis who do not have clinical synovitis?”.
The authors presented a review assessing the most relevant studies investigating the value of US in the prediction of rheumatoid arthritis (RA) development in individuals “at-risk” of RA who have joint symptoms, but no clinical evidence of IA.
The topic is interesting, however I have some minor comments for the Authors:
- In Introduction the authors write: “anti-cyclic citrullinated peptide antibodies (ACPA)”, but in fact the expression should be: s anti-citrullinated peptide/protein antibodies (ACPA). It should be corrected
- What is the origin of “Figure 2. Representative US findings in at-risk individuals with musculoskeletal symptoms but without clinical arthritis.”? Do pictures belong to the Authors or they have been taken from other study? It should be written in the Legend.
Author Response
The authors presented a review assessing the most relevant studies investigating the value of US in the prediction of rheumatoid arthritis (RA) development in individuals “at-risk” of RA who have joint symptoms, but no clinical evidence of IA.
- Thanks to the Reviewer for their kind comments
In Introduction the authors write: “anti-cyclic citrullinated peptide antibodies (ACPA)”, but in fact the expression should be: s anti-citrullinated peptide/protein antibodies (ACPA). It should be corrected
- This has now been changed according to the Reviewers' suggestion
What is the origin of “Figure 2. Representative US findings in at-risk individuals with musculoskeletal symptoms but without clinical arthritis.”? Do pictures belong to the Authors or they have been taken from other study? It should be written in the Legend.
- These are personal images belonging to one of the authors (Andrea Di Matteo). This has now been specified in the Legend of the figure.